# Towards Efficient Luminescent Solar Energy Concentrator Using Cuprorivaite Infrared Phosphor (CaCuSi_4_O_10_)—Effect of Dispersing Method on Photoluminescence Intensity

**DOI:** 10.3390/ma14143952

**Published:** 2021-07-15

**Authors:** Piotr Sobik, Olgierd Jeremiasz, Paweł Nowak, Agata Sala, Bartłomiej Pawłowski, Grażyna Kulesza-Matlak, Anna Sypień, Kazimierz Drabczyk

**Affiliations:** 1Institute of Metallurgy and Materials Science of Polish Academy of Sciences, Reymonta 25 Str., 30-059 Kraków, Poland; piotr.sobik@helioenergia.com (P.S.); olgierd.jeremiasz@helioenergia.com (O.J.); g.kulesza@imim.pl (G.K.-M.); a.sypien@imim.pl (A.S.); 2Helioenergia sp. z o.o., Rybnicka 68 Str., 44-238 Czerwionka-Leszczyny, Poland; pawel.nowak@helioenergia.com (P.N.); agata.sala@helioenergia.com (A.S.); bartlomiej.pawlowski@helioenergia.com (B.P.); 3Institute of Thermal Technology, Silesian University of Technology, Konarskiego 22 Str., 44-100 Gliwice, Poland

**Keywords:** egyptian blue, luminescence solar concentrator, photovoltaic module

## Abstract

Cuprorivaite, also known as Egyptian blue (EB), CaCuSi_4_O_10_, has been utilized as an important blue pigment for thousands of years. It shows a 430–800 nm broad excitation band and an intense 910–920 nm near-infrared (NIR) emission peak at room temperature. The application that motivates the current research is for luminescent solar concentrator (LSC) usage. Current technology for this purpose relies on high near-infrared reflectance. This article addresses the investigation of the relationship between dispersing methods and photoluminescence (PL) intensity. Mechanical grinding methods investigated in the study were: horizontal bead mill, exfoliation and three-roll mill. The initial aim of the study was to verify if the proposed methods do not damage PL. To the surprise of the authors, three-roll mill treatment enhanced PL by nearly 50% without altering the morphology of the powder. An X-ray diffraction study suggested slight alterations in the crystal lattice.

## 1. Introduction

There is a limited number of known luminophores that exhibit near-infrared luminescent properties [1]. Moreover, the available ones may have a limited lifespan and may lose the ability to luminescence due to the action of oxygen, weather conditions, temperature or even light [2]. One of the pigments that exhibits luminescent properties is calcium copper tetrasilicate, CaCuSi_4_O_10_, also known as Egyptian blue. This compound occurs naturally in the form of the cuprorivaite mineral, which was confirmed by X-ray diffraction analysis [3]. This material was produced for decorative purposes in the times of ancient Egypt, in the 3rd millennium BC [4], and was the first synthetic pigment ever produced by humans [5,6,7].

CaCuSi_4_O_10_ consists of layers of linked SiO_4_ tetrahedra and CuO_4_ squares which are separated by calcium cations. The Egyptian blue structure model is shown in Figure 1.

The superior stability of this ceramic phosphor is evidenced by the fact that archaeologists have discovered artifacts decorated with Egyptian blue which still show luminescent abilities after several thousand years. This pigment owes its blue color to the square-planar coordination (D4h) of the CuO_4_ chromophore [8]. This system is controlled by the electrostatic potential created by the lattice over the chromophore, as described by Garcia-Fernandez [8]. Due to the characteristic luminescence spectrum, this material can be distinguished from the similar Han blue in the form of BaCuSi_4_O_10_. The distinction based on photoluminescence was proposed by Pozza et al. [9] who reported the luminescence spectra for both materials. Luminescent photographic techniques in the NIR range thus enable error-free identification and mapping of both ancient pigments [10]. In 2009, a detailed description of the properties of near-infrared luminescence was presented by Accorsi et al. [11]. Calcium copper tetrasilicate shows high quantum efficiency of 10.5% with an emission peak of 910 nm. The excited state lifetime ranges from 107 to 159 µs depending on the degree of fragmentation of the material [12]. Three different electronic transitions related to Cu^2+^ ions were identified for the material: ^2^B1g → ^2^B2g, ^2^Eg and ^2^A_1_g [13].

Egyptian blue is a phosphor which shows the highest quantum efficiency in the NIR range (800–1100 nm). The Stokes shift for this material is as high as 280 nm, which minimizes reabsorption events. Bearing in mind that a silicon solar cell has the ability to absorb in the NIR range for wavelengths around 1000 nm and that the solar spectrum itself is poor in such wavelengths after passing through the atmosphere, this material is a good candidate as a phosphor for applications in luminescent solar energy concentrators [14]. Such a solution was proposed by Li et al. [15], with a condition that it is required to select the matrix so that the refractive index match is consistent, which would eliminate unwanted light scattering. However, it should be noted that EB exhibits anisotropic properties in the case of the refractive index (n_α_ = 1.636, n_β_ = 1.591), and thus the selection of an appropriate matrix remains a challenge. The use of EB in LSCs was also suggested by Berdahl [16], who indicated the possibility of increasing the quantum efficiency of the phosphor.

The quantum efficiency of the phosphors together with the Stokes shift are the features of the phosphors that determine their usefulness in luminescent energy concentrators. Both of these factors have a direct impact on the overall efficiency of the module using luminescence support. As Debije indicates, the optical efficiency of a concentrator depends on the efficiency of the factors, as shown in the following formula [17]:(1)$$\eta_{opt}=\left(1-R\right)P_{TIR}\cdot \eta_{abs}\cdot \eta_{PLQY}\cdot \eta_{Stokes}\cdot \eta_{host}\cdot \eta_{TIR}\cdot \eta_{self}$$
where:

*R*—reflection of solar light from the waveguide surface, *P_TIR_*—the total internal reflection efficiency, *η_abs_*—the solar light fraction absorbed by the dye, *η_PLQY_*—the phosphorus photoluminescent quantum yield, *η_Stokes_*—the energy lost in absorption–emission events, *η_host_*—transport efficiency of the waveguided photons via the matrix, *η_TIR_*—the reflection efficiency of the waveguide due to the smoothness of the waveguide surface, *η_self_*—the transport efficiency of the waveguided photons due to reabsorption of the emitted photons by another luminophore.

Two factors that directly depend on the luminescent material used in the concentrator are the phosphorus photoluminescent quantum yield (*η_PLQY_*) and the energy loss in absorption–emission events (*η_Stokes_*). Thus, stable materials with high quantum efficiency and a Stokes shift minimizing the reabsorption phenomenon are desired.

Synthetic EB is produced in the calcination process at a temperature range of 850–1000 °C. For this purpose, dry substrates of Si, Cu and Ca are mixed with each other and then fired together with the flux. Typical sources are CaCO_3_ for calcium, CuO_3_*Cu(OH)_2_ for copper and SiO_2_ powder for silicon, and the flux concentration does not exceed 5% [16]. An analysis of the methods of obtaining synthetic cuprorivaite was presented by Canti and Heathcote [18], who proposed their own method using the following substrates: black copper oxide (CuO), calcium carbonate (CaCO_3_), silica (SiO_2_) and sodium carbonate as a flux. All the described methods are carried out in an oxidizing atmosphere. However, it should be noted that these methods do not target luminescent properties.

The research on the luminescent properties of EB is most often based on commercially available powders. The procedure for the synthesis of CaCuSi_4_O_10_, taking into account the luminescent properties, was presented by Johnson-McDaniel et al. [19], proposing a variant of the dry synthesis without flux at a temperature of 1025 °C or with the flux at a temperature of 875 °C. At the same time, they present a method of grinding the powder by exfoliation and producing an ink with NIR photoluminescent properties. The important role of washing out the finished powder in HCl solution is also worth mentioning. The experience of Berdahl [16] shows that washing off even commercial powder can significantly increase the luminescence efficiency of the product.

Alternatively, CaCuSi_4_O_10_ can be obtained by the hydrothermal method, however, this method is much more common for variants where the Ca cation is replaced by Sr or Ba [20]. The powders obtained in such a process show a shorter decay time than those made by the solid-state reaction method [16].

Currently, the literature does not report the implementation of a functioning luminescent energy concentrator using Egyptian blue and a photovoltaic cell combined into one system.

The use of Egyptian blue in the form of luminescent centers, however, requires the processing of the raw material in order to obtain a powder of smaller granulation. Such a powder could then be used as a material base for paste or ink, which can be applied by conventional application techniques, including printing techniques. However, it should be noted that powder grinding, depending on the technique, may affect the luminescent properties of Egyptian blue, which will be manifested by change in the quantum efficiency.

## 2. Materials and Methods

Egyptian blue was received in the form of powder which was in the range of 3 to 100 µm in size. Subsequent steps were homogenization and dispersion to prepare it to be incorporated into the matrix material. The aim of the present study was to focus on dispersion methods of EB to check whether dispersion method influences PL intensity.

### 2.1. Sample Preparation

An Egyptian Blue sample was synthesized from 11.98 g of CaCO_3_, 28.76 g of SiO_2_ and 9.52 g of CuO. Reagents were technically pure. CaCuSi_4_O_10_ was synthesized via the dry calcination method. The resulting product was manually treated in a mortar and then washed in 0.3 M HCl solution for 24 h to remove CuO impurities. The powder was then washed out using deionized water and dried in air below 100 °C. This material was a reference material in further studies.

The following dispersion methods were examined: (I) 15 days continuous exfoliation in deionized water with no additional chemistry, (II) horizontal bead mill and (III) three-roll mill. Additionally, three-roll mill dispersion was performed using 2 alternative dispersion media.

The exfoliation process (I) was conducted on 1 g of Egyptian blue powder immersed in 100 mL of demineralized water in a process similar to that proposed by Johnson-McDaniel and Salguero [19]. The solution was mechanically stirred at 400 RPM for 15 days and sampled every 3 days to observe the progress of delamination of the material into smaller flakes. A total of five samples were collected, starting on the third day of the trial.

Samples prepared with a horizontal bead mill (II) were procured using a DISPERMAT^®^ AS rotor–stator dispersion unit from VMA-Getzmann. The process was conducted at 3000 rpm for 4 h.

Dispersions with a three-roll mill (III) were conducted using an Exakt 80 E machine. Each sample was processed 3 times, applying successive gaps between rolls of 20 µm, 15 µm and 10 µm. Each time, the auxiliary medium was added during rolling. The auxiliary medium M1 was commercial compound “80 683” from ETC Sp. z o.o. Ostrowiec Sw. Poland which is a mixture of glycol ether, ethanol and cellulose derivative and the second auxiliary medium M2 was commercial compound “MS-40” from Print-Glass Piotr Sypek, Czestochowa, Poland.

### 2.2. Examination

Digital light microscopy and scanning electron microscopy were applied to check powder morphology. The X-ray diffraction method was used for crystal structure verification.

Digital light microscopy imaging, particle counts and measurements were performed using a Keyence VHX7000 microscope (Keyence Corporation of America, Elmwood Park, NJ, USA).

The particle morphology was examined using an in-lens detector image in the SEM LEO 1530 system (Carl Zeiss AG, Oberkochen, Germany). The investigation was performed at high pressure of 5 × 10^−6^ mbar, 1 kV voltage at different magnifications varying from 1000× to 5000×.

XRD measurements were carried out on a Philips two-stand diffractometer (PW 1710 and PW 1830 (Malvern Panalytical, Malvern, UK)) with the X’Pert system, equipped with the ATC-3 texture goniometer.

Photoluminescence imaging was achieved with an NIR imaging system in the form of a Moravian Instruments G4-16000 CCD camera Mark II with a 4096 (H) × 4096 (V) Full Frame CCD Image Sensor KAF-16803 from ON Semiconductor. The photoluminescent imaging setup was used as shown in Figure 2.

In order to stop the excitation wavelength, an IR bandpass filter was selected. The excitation source was a 65 W electrical power 808 nm laser. In the photoluminescence effect verification, each sample contained the same amount of Egyptian blue.

The last measurement was the determination of photoluminescence quantum yield (PLQY). The powders were measured according to the procedure proposed by de Mello et al. [22]. Egyptian blue, placed in a polytetrafluoroethylene (PTFE) integrating sphere, was excited by semiconductor lasers with a maximum emission at a 656 nm wavelength. The response was measured using a 180 mm diameter integrating sphere made of PTFE and a BWtek Quest X spectrometer with grating covering the 350–1050 nm range and 10 µm slit.

## 3. Results

Digital light microscopy used to compare Egyptian blue powder grains before and after dispersion processes is presented in Figure 3.

Additionally, for each specimen, every grain was measured for its area, major axis, minor axis and perimeter. For morphology, the study was based on derived coefficients of size (2), compactness (3), regularity (4) and circuitry (5), as follows:(2)$$size=\frac{major\;axis+minor\;axis}{2},$$
(3)$$compactness=\frac{perimeter^{2}}{4\Pi \;\mathrm{area}},$$
(4)$$Lp2=\frac{major\;axis}{perimeter},$$
(5)$$W9=\frac{2\;\sqrt{\Pi \;area}}{perimeter}.$$

Compactness was used as a measure of shape regularity. For this coefficient, objects which are close to a circle take a minimum value of 1. Objects that have complicated, irregular boundaries are defined with larger compactness. In the case of regularity coefficient objects having extended irregular boundaries, they have a small Lp2 value. Circuitry coefficient W9 approaches a value of 1 for shapes close to a circle. The distribution of the chosen coefficients is presented in Figure 4.

Scanning electron microscopy images are presented in Figure 5, whereas photoluminescence images of Egyptian blue with respect to different dispersion operations are presented in Figure 6.

XRD measurements detected two significant phases in the form of cuprorivaite and cristobalite. The X-ray phase analysis was performed using the HighScore Plus program and the COD crystallographic database. Figure 7 presents the XRD pattern of Egyptian blue after horizontal bead mill treatment.

The intensity distribution (A1, A2, A3 and A4) may indicate the presence of other phases in the sample at the limit of detection of the applied research method. Lattice parameters with respect to dispersion method are presented below (Table 1).

Finally, photoluminescence quantum yield has been determined for each of the powders. The obtained efficiencies are presented in Table 2. In addition, a measurement was performed for a commercial Kremer’s Egyptian blue powder. In order to eliminate the influence of the absorption of radiation emitted by the Egyptian blue by water vapor and O_2_, the measurement was carried out in a nitrogen atmosphere.

Since noticeable differences in PLQY were noted for Egyptian blue grains resulting from horizontal bead mill and three-roll mill processes, more detailed imaging of them was performed. Figure 8 presents comparisons of grains achieved in those two methods with respect to the reference powder.

The images indicated a difference in grain shapes procured in different dispersion methods. Powder processed with a three-roll mill gave more flake-like shapes whereas the horizontal bead mill dispersion resulted in more spheroidal grains.

## 4. Discussion

In order to use Egyptian blue to make an effective luminescence solar concentrator, it is necessary to produce a fine powder that could be used as a material base for production of printing pastes, inks and paints. This can be achieved by material grinding and milling which can affect the quantum efficiency of a dispersed powder.

Light microscopy followed by detailed analysis of size and morphology showed that the three-roll mill treatment resulted in no significant changes in size and morphology, while these samples exhibited the highest photoluminescence enhancement. The exfoliation process mostly alters the size and morphology but it is followed by a strong photoluminescence decrease. Bead mill treatment creates the largest crystallites in size, while the PL image remains unchanged. SEM imaging shows that three-roll mill samples possess the most sharp edges and a layered structure of crystals can be observed. Therefore, it is concluded that for the presented dispersion methods, another factor besides morphology and size could be responsible for the phenomenon of PLQY change.

The XRD measurement showed the presence of two major phases. The cristobalite accompanying the Egyptian blue is the result of a high-temperature powder manufacturing process and can be minimized in a flux-assisted synthesis process [23]. XRD verification may suggest that slight changes in crystal lattice constants are created during three-roll mill treatment. However, those changes are questionable as they are at the limit of equipment sensitivity. Three-roll mill treatment resulted in a significant PL enhancement which must be explained with further studies.

As explained by Chen et al. [24], applying biaxial tensile strain can linearly tune the magnitude of energy band gaps but the direct gap feature remains.

The obtained results of PLQY significantly differ from the highest reported quantum efficiency at the level of Φ_EM_ = 10.5% as presented by Accorsi et al. [11]. The determined level of PLQY for powders subjected to various methods of mechanical treatment is, however, strongly related to the treatment. The decrease in PLQY may occur due to the presence of cristobalite powder. It should also be noted that the commercial powder represented by Kremer Egyptian blue used extensively in the work showed a much lower efficiency than the record-high efficiency, which may explain why there is no functional energy concentrator based on this material to date. At the same time, the authors of the study recommend absolute verification of the quantum efficiency of the obtained powders after each treatment.

## Figures and Tables

**Figure 1 materials-14-03952-f001:**
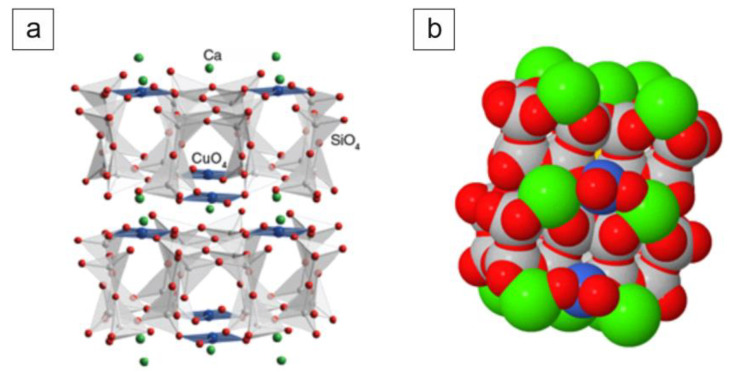
Spatial model of Egyptian blue (calcium copper tetrasilicate, CaCuSi_4_O_10_), (**a**) model with highlighted silicon SiO_4_ sites and copper CuO_4_ sites, (**b**) model with visible Ca cations, source ChemTube3D by Nick Greeves, licensed under CC BY-NC-SA 2.0 UK.

**Figure 2 materials-14-03952-f002:**
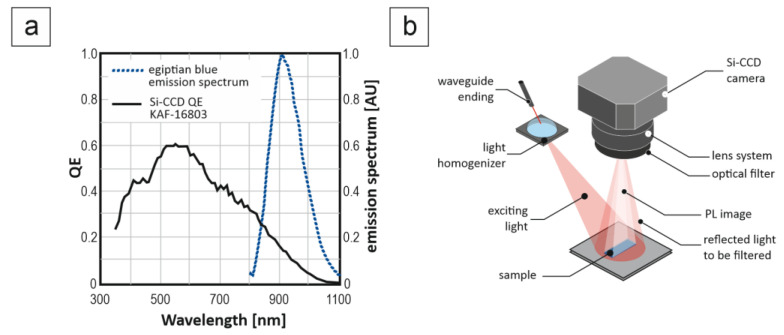
Egyptian blue photoluminescent imaging: (**a**) relation between camera detection spectrum and Egyptian blue emission spectrum (based on Semiconductor Components Industries KAF−16803/D datasheet [21] and Errington [1]), (**b**) measurement system for photoluminescence imaging.

**Figure 3 materials-14-03952-f003:**
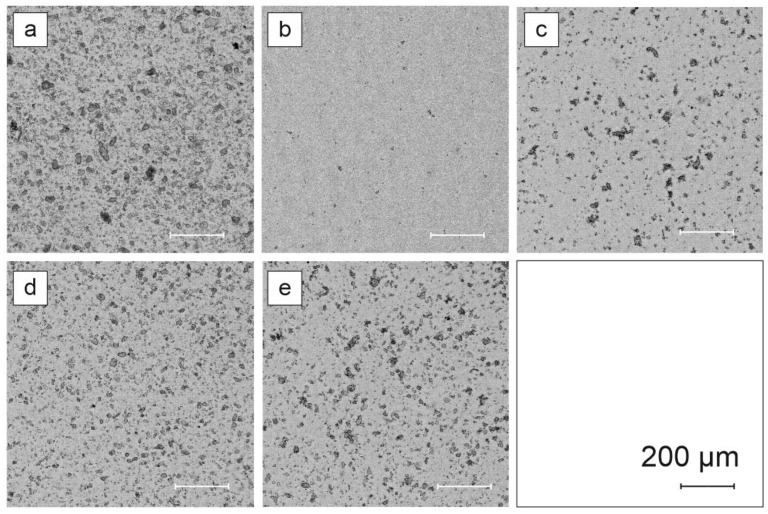
Digital light microscopy images of EB samples after different dispersion operations: (**a**) reference powder, (**b**) 15 days of exfoliation (I), (**c**) horizontal bead mill (II), (**d**) three-roll mill (III) with medium M1, (**e**) three-roll mill (III) with medium M2.

**Figure 4 materials-14-03952-f004:**
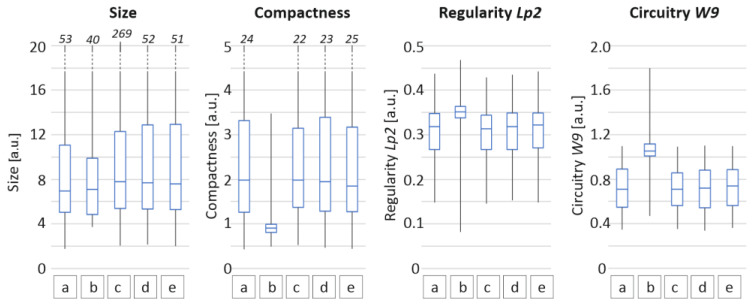
Coefficients of size, compactness, regularity and circuitry for Egyptian blue powders obtained after different dispersion operations: (a) reference powder, (b) 15 days of exfoliation (I), (c) horizontal bead mill (II), (d) three-roll mill (III) with medium M1, (e) three-roll mill (III) with medium M2.

**Figure 5 materials-14-03952-f005:**
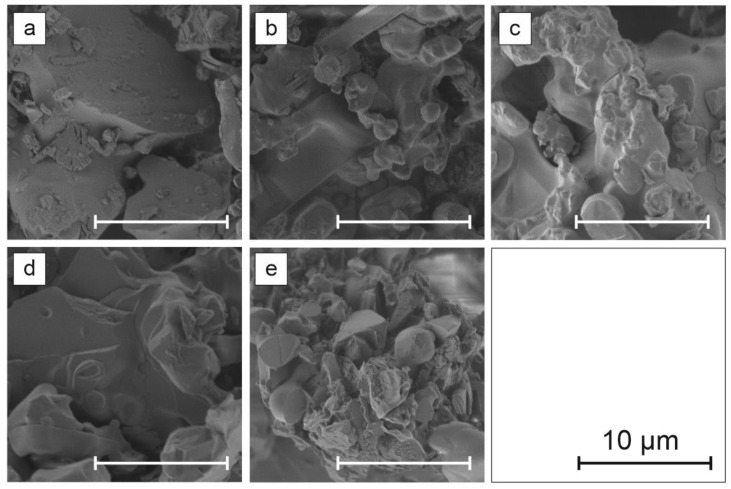
SEM images of EB samples after different dispersion operations: (**a**) reference powder, (**b**) 15 days of exfoliation (I), (**c**) horizontal bead mill (II), (**d**) three-roll mill (III) with medium M1, (**e**) three-roll mill (III) with medium M2.

**Figure 6 materials-14-03952-f006:**
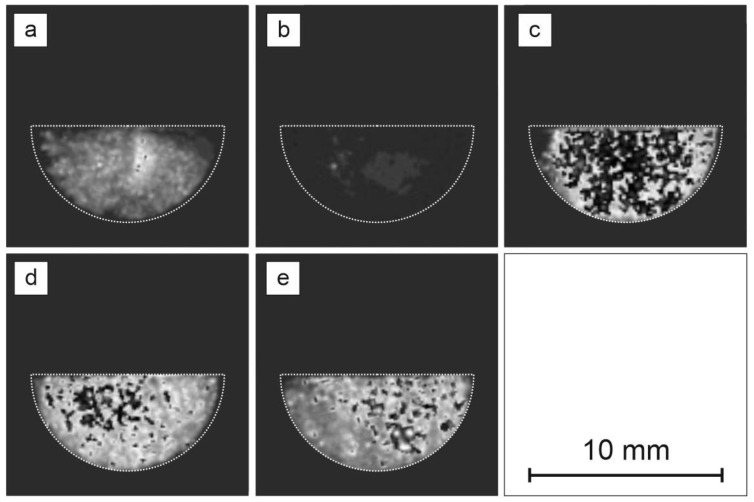
Photoluminescence images of Egyptian blue after different dispersion operations (**a**) reference powder, (**b**) 15 days of exfoliation (I), (**c**) horizontal bead mill (II), (**d**) three-roll mill (III) with medium M1, (**e**) three-roll mill (III) with medium M2.

**Figure 7 materials-14-03952-f007:**
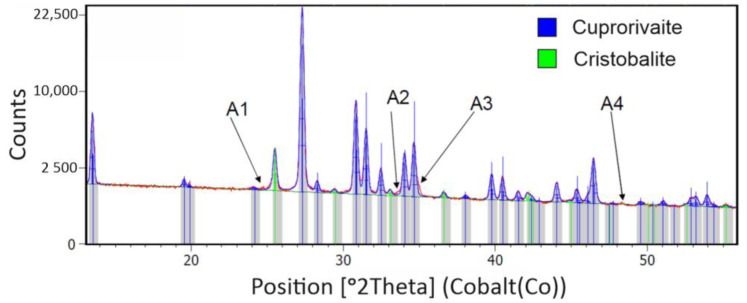
XRD pattern of Egyptian blue after horizontal bead mill treatment with two significant phases detected: cuprorivaite and cristobalite. Arrows A1–A4 indicate disturbances in the intensity distribution.

**Figure 8 materials-14-03952-f008:**
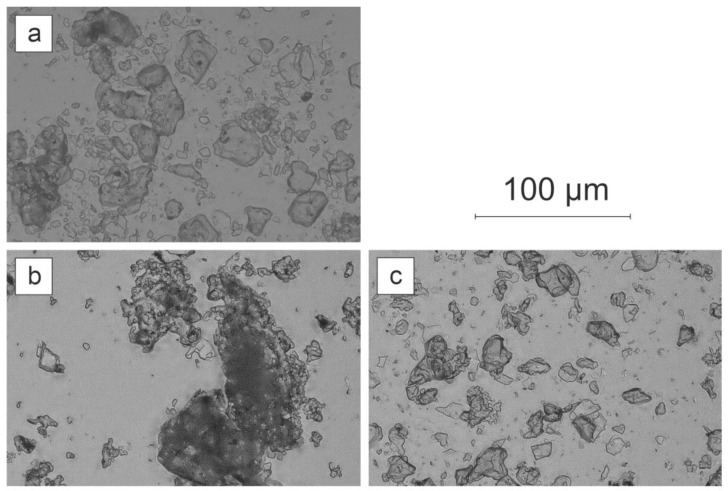
Digital light microscopy images of EB grains after different dispersion operations: (**a**) reference powder, (**b**) horizontal bead mill (II), (**c**) three-roll mill (III) with medium M1.

**Table 1 materials-14-03952-t001:** Cuprorivaite and cristobalite lattice parameters and relative phase amount with respect to dispersion method.

Specimen	Cuprorivaite			Cristobalite		
	a [Å]	c [Å]	Relative Phase Amounts in Mass [%]	a [Å]	c [Å]	Relative Phase Amounts in Mass [%]
Reference	7.3021	15.1235	90.1	4.9744	6.9340	9.9
Exfoliation	7.3008	15.1204	88.02	4.9717	6.9337	11.98
Horizontal bead mill	7.3009	15.1210	89.4	4.9738	6.9291	10.6
Three-roll mill M1	7.3023	15.1247	91.9	4.9725	6.9451	8.1
Three-roll mill M2	7.3021	15.1245	89.3	4.9729	6.9425	10.7

**Table 2 materials-14-03952-t002:** Photoluminescence quantum yield with respect to dispersion method.

Specimen	PLQY [%]
Reference	0.50
15 days of exfoliation	undetected
Horizontal bead mill	0.38
Three-roll mill M1	0.74
Three-roll mill M2	0.71
Egyptian blue by Kremer	1.36

## Data Availability

Not applicable.

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
