# Peer review of "Towards Efficient Luminescent Solar Energy Concentrator Using Cuprorivaite Infrared Phosphor (CaCuSi_4_O_10_)—Effect of Dispersing Method on Photoluminescence Intensity"

_materials, 2021, doi:10.3390/ma14143952_

Round 1

Reviewer 1 Report

Kazimierz Drabczyk et. al reported work " Towards efficient luminescent solar energy concentrator using 2 cuprorivaite infrared phosphor (CaCuSi4O10) - effect of dispersing method on photo-luminescence intensity" is interesting can be acceptable with major revisions as specified below. 

1) Characterisation of dispersed EB samples only SEM does not provide composition on surface which decides intensity of photo-luminescence of solid materials. Therefore it is highly recommended to provide SEM-EDX analysis along with XPS data.

2)  As the title of the article clearly states that, towards luminescent solar energy concentrator, plausible reason behind the increment of PL intensity need to be described briefly. 

Reviewer 2 Report

The manuscript describes the “Towards efficient luminescent solar energy concentrator using cuprorivaite infrared phosphor (CaCuSi4O10)-effect of dispersing method on photoluminescence intensity”. The structural, morphological, and electrochemical properties of the films were studied and compared with their binary counterparts. After careful evaluation of the paper, I recommend publication subject to a major revision in the following aspects,

The author described the “X-ray diffraction study suggest slight alterations in crystal lattice” in the paper, but where is XRD figure? And the author determined the lattice parameters?

The SEM and TEM images are not clear.

In this current form, there are some mistakes in the main text and your figures. Please correct them and polish your language carefully.

Please correct some typos.

Compared with previous literature, what are the novelty of your work? Please highlight them.

In your XRD results, the XRD patterns are not showing. Please give a Figure.

Another should revise the Discussion part.  

Reviewer 3 Report

This paper considers the influence of the dispersion method on photoluminescence intensity of the mineral pigment cuprorivaite proposed for application as a near-IR luminophore for luminescent solar energy concentrators. The present study is limited by comparison of the morphology and photoluminescence quantum yield of the mineral samples after different methods of mechanical treatment, so it can hardly be of interest for a wide readership of the journal. However, if the editors consider it suitable for publication in Materials, the following modifications are required:

  1. The authors just describe a phenomenon of photoluminescence enhancement in a number of samples after mechanical treatment, but there are no suggestions about the possible physical and chemical mechanisms of the effect observed.
  2. Optical and SEM images indicate a rather high degree of polydispersity in the samples both before and after dispersion with the corresponding scattering in morphometric data (except for those after exfoliation), so the dispersion can be considered ineffective. The above fact makes the following comparison and the final conclusions rather doubtful. For more detailed study, it is recommended to perform separate experiments (PLQY determination and XRD measurements) for different particle fractions with a narrow particle size distribution in order to establish whether the particle size or its crystal structure effect the resulting quantum yield.
  3. The presence of different impurities can significantly modify luminescent properties of the luminophore, so this problem should also be treated more carefully considering all the factors which can influence the sample during different dispersion procedures.
  4. All the quantum yield values given in table 2 are rather low, and the possibility of application of the mineral powder with such parameters in luminescent solar energy concentrators seems to be questionable.
  5. The English language of the manuscript is very poor, so it requires proofreading and correction of numerous spelling and grammatical errors before publication.
  6. The reference list is very scarce. It is also strongly recommended to remove the reference from the abstract, since it would be impossible to access it in the absence of the full text.
  7. The authors should specify the source of Figure 1 with the appropriate reference and permission.

Round 2

Reviewer 1 Report

Revised manuscript can be acceptable in current form.

Reviewer 2 Report

The author revised paper in the good order, now from my side paper can be accepted.